# Interneurons and oligodendrocyte progenitors form a structured synaptic network in the developing neocortex

David Orduz[1,2†], Paloma P Maldonado[1,2†‡], Maddalena Balia[1,2], Mateo Vélez-Fort[1,2§], Vincent de Sars[2,3], Yuchio Yanagawa[4], Valentina Emiliani[2,3], Maria Cecilia Angulo[1,2*]

[1]INSERM U1128, Paris, France; [2]Université Paris Descartes, Sorbonne Paris Cité, Paris, France; [3]CNRS UMR8250, Paris, France; [4]Department of Genetic and Behavioral Neuroscience, Gunma University Graduate School of Medicine, Maebashi, Japan

**Abstract** NG2 cells, oligodendrocyte progenitors, receive a major synaptic input from interneurons in the developing neocortex. It is presumed that these precursors integrate cortical networks where they act as sensors of neuronal activity. We show that NG2 cells of the developing somatosensory cortex form a transient and structured synaptic network with interneurons that follows its own rules of connectivity. Fast-spiking interneurons, highly connected to NG2 cells, target proximal subcellular domains containing GABA$_A$ receptors with $\gamma$2 subunits. Conversely, non-fast-spiking interneurons, poorly connected with these progenitors, target distal sites lacking this subunit. In the network, interneuron-NG2 cell connectivity maps exhibit a local spatial arrangement reflecting innervation only by the nearest interneurons. This microcircuit architecture shows a connectivity peak at PN10, coinciding with a switch to massive oligodendrocyte differentiation. Hence, GABAergic innervation of NG2 cells is temporally and spatially regulated from the subcellular to the network level in coordination with the onset of oligodendrogenesis.

## Introduction

The discovery of bona fide synapses formed on non-neuronal NG2-expressing cells (*Bergles et al., 2000*), the progenitors of myelinating oligodendrocytes, has challenged the dogma that synapses are a unique feature of neurons in the central nervous system. Since then, the existence of functional synapses between neurons and NG2 cells is recognized as a major physiological feature of these cells throughout the brain (*Maldonado and Angulo, 2014*). In the somatosensory cortex, NG2 cells receive a major synaptic input from local GABAergic interneurons that disappears after the second postnatal (PN) week (*Vélez-Fort et al., 2010*; *Balia et al., 2015*). Cortical NG2 cells are, therefore, believed to be transiently embedded in GABAergic microcircuits at a period known to undergo oligodendrocyte differentiation in the neocortex (*Baracskay et al., 2002*). However, while the connectivity patterns between neocortical interneurons and their neuronal partners begin to be elucidated (*Fino and Yuste, 2011*; *Pfeffer et al., 2013*), the rules governing the GABAergic innervation of NG2 cells in the network are elusive.

Cortical GABAergic interneurons are one of the most heterogeneous populations of neurons in the brain (*Cauli et al., 1997*; *Petilla Interneuron Nomenclature Group et al., 2008*). Their diversity has been a matter of intense investigation for several decades and is known to impact synaptic signaling and computational capacities of neuronal networks (*Klausberger and Somogyi, 2008*; *Fishell and Rudy, 2011*). Different types of interneurons target specific subcellular compartments of their postsynaptic neuron. Such compartmentalization creates a specific distribution of channels, receptors,

---

*For correspondence: maria-cecilia.angulo@parisdescartes.fr

†These authors contributed equally to this work

**Present address:** ‡Netherlands Institute for Neuroscience, The Royal Academy of Arts and Sciences, Amsterdam, Netherlands; §Sainsbury Wellcome Centre for Neural Circuits and Behaviour, University College London, London, UK

**Competing interests:** The authors declare that no competing interests exist.

**eLife digest** Neurons are outnumbered in the brain by cells called glial cells. The brain contains various types of glial cells that perform a range of different jobs, including the supply of nutrients and the removal of dead neurons. The role of glial cells called oligodendrocytes is to produce a material called myelin: this is an electrical insulator that, when wrapped around a neuron, increases the speed at which electrical impulses can travel through the nervous system.

Neurons communicate with one another through specialized junctions called synapses, and at one time it was thought that only neurons could form synapses in the brain. However, this view had to be revised when researchers discovered synapses between neurons and glial cells called NG2 cells, which go on to become oligodendrocytes. These neuron-NG2 cell synapses have a lot in common with neuron–neuron synapses, but much less is known about them.

Orduz, Maldonado et al. have now examined these synapses in unprecedented detail by analyzing individual synapses between a type of neuron called an interneuron and an NG2 cell in mice aged only a few weeks. Interneurons can be divided into two major classes based on how quickly they fire, and Orduz, Maldonado et al. show that both types of interneuron form synapses with NG2 cells. However, these two types of interneuron establish synapses on different parts of the NG2 cell, and these synapses involve different receptor proteins.

Together, the synapses give rise to a local interneuron-NG2 cell network that reaches a peak of activity roughly two weeks after birth, after which the network is disassembled. This period of peak activity is accompanied by a sudden increase in the maturation of NG2 cells into oligodendrocytes. Further experiments are needed to test the possibility that activity in the interneuron-NG2 cell network acts as the trigger for the NG2 cells to turn into oligodendrocytes, which then supply myelin for the developing brain.

and signaling mechanisms and allows for an effective regulation of synaptic integration, plasticity, and spiking (*Huang et al., 2007*). For instance, it has been observed that the localization of different $GABA_A$ receptors ($GABA_ARs$) in neocortical pyramidal neurons is input-specific since presynaptic parvalbumin (PV)-positive, fast-spiking cells innervate proximal postsynaptic sites with $GABA_ARs$-containing α1 subunits, whereas bitufted interneurons contact postsynaptic sites with $GABA_ARs$-containing α5 subunits (*Ali and Thomson, 2008*). At a higher level, the connectivity patterns of neocortical interneurons in the network also appear to be highly specific (*Pfeffer et al., 2013*). PV-positive interneurons strongly inhibit one another but provide little inhibition to other subtypes of interneurons, whereas somatostatin-positive interneurons strongly inhibit all other interneurons but are poorly interconnected with each other (*Pfeffer et al., 2013*). Despite the existence of specific connectivity patterns among interneurons, this heterogeneous cell population carves out unspecific and dense connections with pyramidal cells (*Fino and Yuste, 2011*; *Packer et al., 2014*). Hence, the connectivity of interneurons cannot be generalized and categorized in a simple way.

Is interneuron-NG2 cell connectivity governed by any specific rule? Our knowledge of NG2 cell synaptic physiology and connectivity is still very limited because it derives exclusively from studies based on spontaneous synaptic activity or on averaged synaptic currents generated by the stimulation of unidentified neurons. No information exists on the identity of presynaptic inputs impinging on NG2 cells, the dynamics of their individual synapses, and their microcircuit architectures.

Here, we investigate the properties of unitary interneuron-NG2 cell connections during the critical period of NG2 cell differentiation in the somatosensory cortex of Slc32a1-Venus;Cspg4-DsRed transgenic mice (hereafter called VGAT-Venus;NG2-DsRed mice). By combining immunohistochemistry, paired recordings, and holographic photolysis for circuit mapping, our results reveal that interneuron-NG2 cell connections in the developing neocortex form a transient and organized local network that is functional only during the most critical days of cortical oligodendrogenesis. A local microcircuit architecture with interneuron-NG2 cell intersomatic distances never exceeding 70 μm is accompanied with a specific subcellular arrangement of inputs from fast-spiking interneurons (FSIs) and from non-fast-spiking interneurons (NFSIs). These two classes of interneurons target different segregated postsynaptic domains containing distinct $GABA_ARs$. In conclusion, these progenitors form

their own structured network with interneurons whose properties are temporally and spatially regulated in concordance with the onset of oligodendrocyte differentiation process.

## Results

### FSIs are highly connected to NG2 cells

To test whether NG2 cells are wired by interneurons in a specific manner, we searched for presynaptic and postsynaptic principles governing individual interneuron-NG2 cell synapses. We performed paired recordings between layer V Venus[+] interneurons and DsRed[+] NG2 cells in acute somatosensory cortical slices of VGAT-Venus;NG2-DsRed mice from PN8 to PN13 (*Figure 1A—figure supplement 1A*). Paired recordings allowed us to characterize the action potential firing behavior of the interneuron, the characteristic conductance profile of the NG2 cell, and the specific synaptic properties of the connection (*Figure 1A—figure supplement 1B–G*; see 'Materials and methods'). In 38 out of 147 pairs, action currents elicited in presynaptic interneurons induced inward postsynaptic currents (PSCs) sensitive to the $GABA_A$R antagonist SR95531 in NG2 cells recorded with a CsCl-based intracellular solution (*Figure 1A—figure supplement 1D*). All unitary connections displayed currents with small amplitudes and showed paired-pulse depression without recovery within 250 ms (*Figure 1A—figure supplement 1D–G*).

FSIs and NFSIs can be distinguished by their firing properties even if they have not attained maturity at this developmental stage (*Daw et al., 2007*). To investigate the identity of recorded interneurons in all tested pairs, we analyzed nineteen different electrophysiological parameters in current-clamp mode (*Figure 1B,D*; *Table 1*). FSIs were primarily distinguished from NFSIs by their narrow action potential waveforms with profound after-hyperpolarizations (AHPs), a negligible spike broadening and spike amplitude reduction during trains (*Figure 1B,D,E*) (*Cauli et al., 1997*; *Daw et al., 2007*). Other seven parameters were also statistically different between FSIs and NFSIs and clearly separated these neurons in two distinct groups (*Table 1*). The identity of FSIs was further confirmed by the expression of PV in biocytin-labeled interneurons, a reliable marker for this cell class that was absent in NFSIs (*Figure 1C—figure supplement 2*). As expected for these two classes of interneurons, FSIs appeared as a relatively homogeneous population with restricted distributions of the main discriminative parameters, whereas NFSIs encompassed different subtypes as revealed by the large distribution of their electrophysiological values (*Figure 1E—figure supplement 3A*) (*Cauli et al., 1997*).

The amount of tested FSIs and NFSIs was in agreement with those of PV[+]/Venus[+] and PV[−]/Venus[+]interneurons in the transgenic mouse, respectively, with NFSIs being more abundant (*Figure 1F,G*). However, the proportion of connected FSIs was high compared to the FSI abundance in tested pairs (*Figure 1G*) or within the population of tested FSIs (*Figure 1H*). Conversely, the proportion of connected NFSIs was low compared to the NFSI abundance in tested pairs (*Figure 1G*) or within the population of tested NFSIs (*Figure 1H*). These data suggest that connection probabilities of FSIs and NFSIs are different. To determine whether differences in the connection probabilities for FSIs (p = 0.43) and NFSIs (p = 0.21) did not arise by chance, we modeled each of the two data sets as observations from two binomial distributions. The estimated connection probabilities at 90% confidence intervals for FSIs and NFSIs were 0.30–0.57 and 0.15–0.27, respectively (*Brown et al., 2001*; *Supplementary file 1*). If the two populations have the same connection probabilities, we expect non-overlapping intervals in at most 1% of the cases (0.1 × 0.1). No overlap was observed between these two intervals, indicating that FSIs and NFSIs have different connection probabilities (p < 0.01). We obtained a similar result using a chi-square test (Pearson's chi-square of 6.93; p < 0.01). In another hand, it is noteworthy that no significant differences of main discriminative electrophysiological parameters were found between tested and connected NFSIs, indicating that NFSIs innervating NG2 cells constitute a heterogeneous population of interneurons as observed in the tested population (*Figure 1—figure supplement 3A,B*). Altogether, these results indicate that NFSIs are by far more abundant and diverse but poorly connected to NG2 cells, whereas FSIs are less abundant but highly connected to NG2 cells in the developing GABAergic network.

Finally, we previously demonstrated that NG2 cells receive a transient GABAergic synaptic input from interneurons that disappears after the second PN week (*Vélez-Fort et al., 2010*; *Balia et al., 2015*). We, thus, tested how this transient connectivity occurs for FSI and NFSI separately. *Figure 1I*

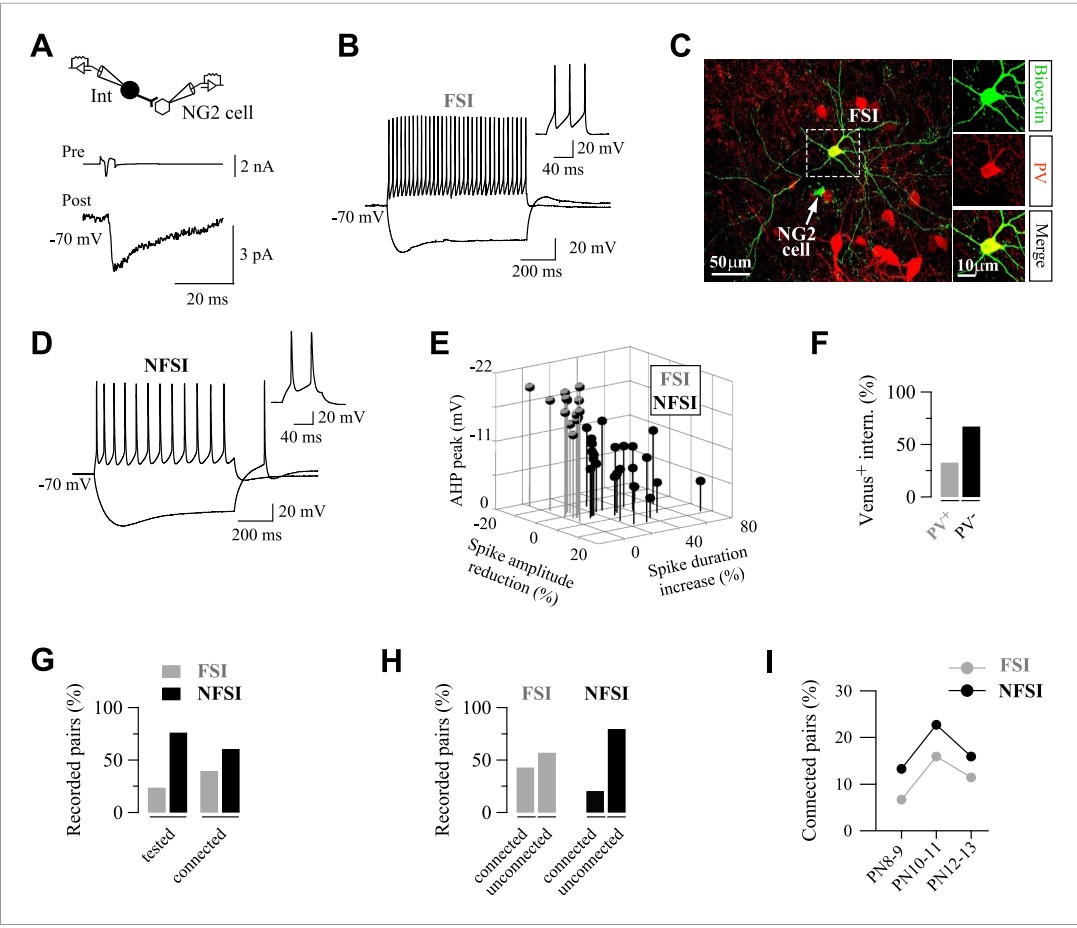

**Figure 1**. FSIs are highly connected to NG2 cells. (**A**) Paired recording between a presynaptic fast-spiking interneuron (FSI) and a NG2 cell. An action current evoked in the presynaptic interneuron (upper trace) elicits PSCs recorded in the NG2 cell (bottom trace; average of 200 traces). (**B**, **D**) Current-clamp recordings of the FSI recorded in **A** (**B**), and a non-fast-spiking interneurons (NFSI) (**D**) connected to a NG2 cell during injections of −150 pA and 200 pA. Note differences on spike properties between the two cells (insets, right). (**C**) The connected FSI was loaded with biocytin and was immunoreactive for PV (stacks of 17 Z-sections; each 2 μm). (**E**) 3D plot of the three major electrophysiological parameters distinguishing connected FSIs (gray) from NFSIs (black). (**F**) Histograms showing the fraction of Venus+ interneurons that are PV+ (n = 6 mice). (**G**) Percentages of tested and connected FSI (gray) and NFSI (black) with respect to all tested interneurons. (**H**) Percentages of connected and unconnected FSIs (gray) and NFSIs (black) with respect to each group of interneurons separately. (**I**) Connection probability of FSI and NFSI as a function of three postnatal stages (45, 44, and 44 tested pairs at PN8-9, PN10-11, and P12-13, respectively).

The following figure supplements are available for figure 1:

**Figure supplement 1**. Paired recordings in double VGAT-Venus;NG2-DsRed transgenic mouse.

**Figure supplement 2**. PV marker is expressed in electrophysiologically identified FSIs, but not in NFSIs.

**Figure supplement 3**. Cumulative distributions for the three main electrophysiological parameters used to differentiate FSIs from NFSIs.

illustrates the connection probability of these interneurons at three different postnatal stages. A transient peak of connectivity occurred at PN10-11 for both FSI and NFSI (*Figure 1I*), indicating that the connectivity of these two classes of interneurons is not differently affected by the stage of development.

**Table 1**. Electrophysiological properties of Venus$^+$ FSI and NFSI

| Parameter | FSI (n = 27) | NFSI (n = 105) | p | Comparison |
|---|---|---|---|---|
| $F_{total}$ (Hz) | 28.32 ± 2.73 | 24.18 ± 0.93 | NS | – |
| $F_{init}$ (Hz) | 37.12 ± 3.89 | 45.29 ± 2.3 | NS | – |
| $F_{200}$ (Hz) | 29.55 ± 2.73 | 24.48 ± 0.89 | NS | – |
| $F_{final}$ (Hz) | 26.36 ± 2.72 | 21.11 ± 0.83 | NS | – |
| Early accommodation | 14.14 ± 3.61 | 39.33 ± 1.85 | <0.0001 | FSI < NFSI |
| Late accommodation | 11.43 ± 3.01 | 8.87 ± 0.68 | NS | – |
| Threshold (mV) | −35.68 ± 0.64 | −36.31 ± 0.47 | NS | – |
| First spike amplitude (mV) | 70.73 ± 2.33 | 70.61 ± 1.09 | NS | – |
| Second spike amplitude (mV) | 70.55 ± 2.08 | 65.51 ± 1.3 | NS | – |
| **Spike amplitude reduction** | **−0.24 ± 0.94** | **7.59 ± 1.14** | **<0.0001** | **FSI < NFSI** |
| First spike duration (ms) | 1.37 ± 0.1 | 2.41 ± 0.1 | <0.0001 | FSI < NFSI |
| Second spike duration (ms) | 1.46 ± 0.11 | 3.83 ± 0.22 | <0.0001 | FSI < NFSI |
| **Spike duration increase** | **6.79 ± 0.75** | **55.06 ± 5.92** | **<0.0001** | **FSI < NFSI** |
| **AHP (mV)** | **−15.89 ± 0.61** | **−8.43 ± 0.4** | **<0.0001** | **FSI > NFSI** |
| AHP width (ms) | 25.57 ± 1.61 | 21.53 ± 1.05 | NS | – |
| Peak to AHP trough (ms) | 7.18 ± 0.92 | 13.85 ± 0.88 | <0.0001 | FSI < NFSI |
| AP-depolarizing slope (mV/ms) | 207.34 ± 12.86 | 140.45 ± 5.63 | <0.0001 | FSI > NFSI |
| AP-hyperpolarizing slope (mV/ms) | −67.38 ± 4.35 | −32.81 ± 1.36 | <0.0001 | FSI > NFSI |
| $R_{in}$ (MΩ) | 208.22 ± 17.97 | 399.96 ± 19.75 | <0.0001 | FSI < NFSI |

For the identification of FSI and NFSI by firing properties, we first analyzed spike-frequencies in Venus$^+$ interneurons during suprathreshold pulses in current clamp configuration from −70 mV (200 pA, 800–1000 ms). Firing frequency was calculated for the entire pulse as the number of spikes divided by the pulse duration ($F_{total}$). Three instantaneous discharge frequencies were also calculated: (1) between the first pair of spikes ($F_{initial}$); (2) at 200 ms from the beginning of the pulse ($F_{200}$); and (3) at the end of the pulse ($F_{final}$). These values were used to quantify both early and late accommodations in accordance with the following formulas: ($F_{initial} - F_{200}/F_{initial}$) and ($F_{200} - F_{final}/F_{initial}$), respectively. We also dissected the spike morphology from action potentials elicited by 80-ms suprathreshold pulses from −70 mV (150–200 pA). From these recordings, the spike threshold corresponded to the voltage at which the derivative of the AP (dV/dt) experienced a twofold increase. The amplitudes, the first and the second AP ($A_1$ and $A_2$), were calculated from the threshold to peak. Their duration ($D_1$ and $D_2$) corresponded to the full-width at half maximum (FWHM) from a Gaussian fit of the depolarized face of the AP immediately after the threshold. Both amplitude reduction and duration increase were calculated by the formulas $A_1 - A_2/A_1$ and $D_2 - D_1/D_1$, respectively. The amplitude of the after-hyperpolarization (AHP) was calculated as the difference between the threshold and the peak of the fast hyperpolarization. We also estimated the AHP width as the FWHM and the latency of AP peak to AHP trough. We extracted the positive and negative peaks from the derivative of the AP waveform to quantify the maximal speed excursion of the membrane voltage during both depolarizing and hyperpolarizing faces of the AP. Finally, the input resistance ($R_{in}$) was measured in current clamp by applying hyperpolarizing pulses from −60 mV (−200 pA). The three major parameters to differentiate FSI from NFSI appeared in bold. Note that seven other parameters also easily differentiate these interneurons. NS: no significant difference.

## FSIs and NFSIs target specific segregated subcellular domains

We next investigated whether FSI-NG2 cell connections could be distinguished from NFSI-NG2 cell connections by their synaptic properties. Paired-pulse ratios, PSC1 and PSC2 amplitudes with or without failures, coefficients of variation, and probabilities of response were not significantly different between FSI-NG2 cell and NFSI-NG2 cell connections, suggesting that presynaptic mechanisms of GABA release did not participate in the selectivity for presynaptic inputs (*Figure 2—figure supplement 1A–E*). However, we observed a significant difference in current kinetics with rise and decay times faster for unitary FSI-NG2 cell connections (*Figure 2A,B*). These kinetic differences could result from two main causes: (1) FSI-NG2 cell and NFSI-NG2 cell synapses are distributed at different

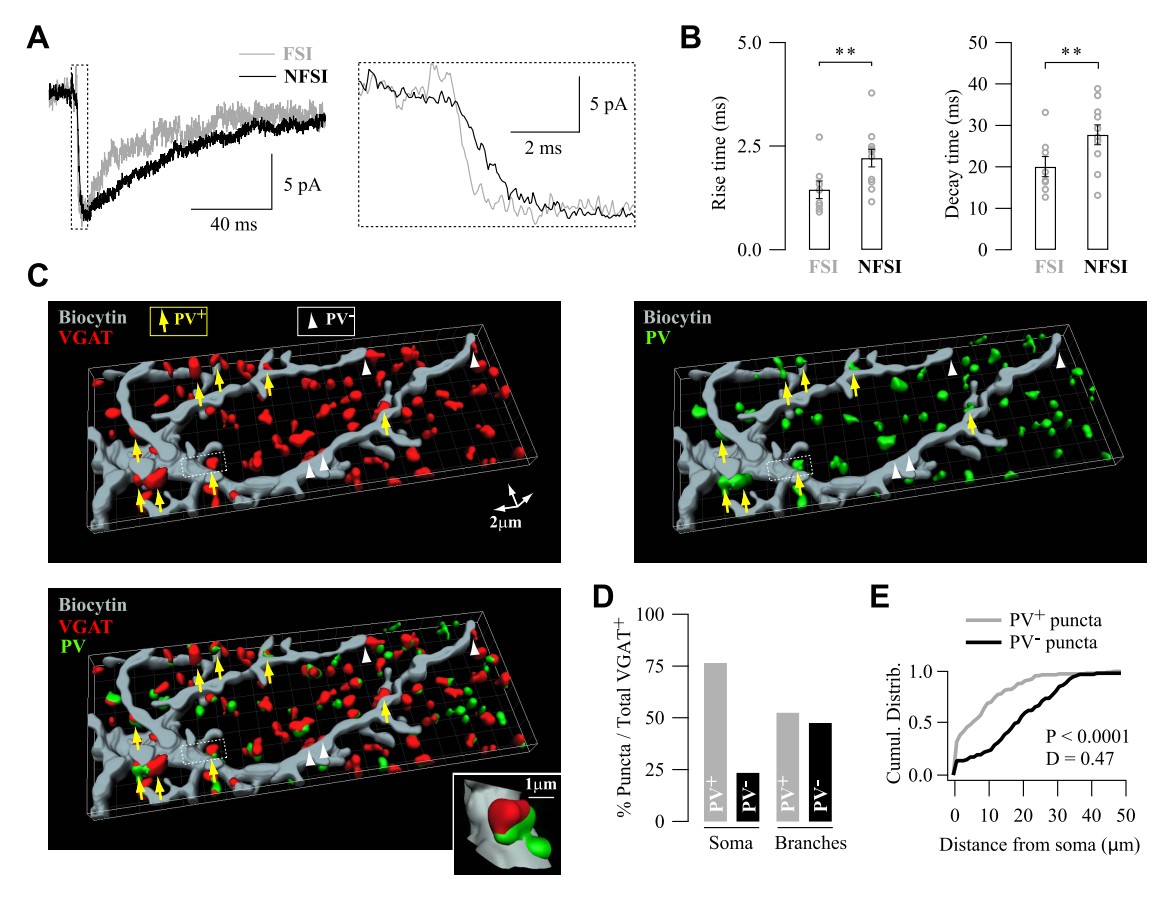

**Figure 2**. Specific subcellular distribution of FSI and NFSI synaptic contacts and postsynaptic GABA$_A$Rs on NG2 cells. (**A**) Different kinetics of unitary postsynaptic NG2 cell currents depends on the identity of the presynaptic interneuron. Superimposed postsynaptic currents evoked in a FSI-NG2 cell connection (gray) and a NFSI-NG2 cell connection (black). Note the faster rise (inset, right) and decay times for the FSI-NG2 cell connection ($t_{10-90}$ = 0.91 ms and $\tau$ = 12.7 ms for the FSI-NG2 cell connection and 2.25 ms and 23.2 ms for the NFSI-NG2 cell connection). Only traces showing a response in the first presynaptic action current were averaged. (**B**) Histogram comparing rise (left) and decay times (right) for different FSI-NG2 cell and NFSI-NG2 cell connections. (**C**) 3D reconstruction of a biocytin-loaded NG2 cell (gray), VGAT (red), and PV (green) labeling at PN10 (see original immunostainings in *Figure 2—figure supplement 2A*). VGAT$^+$/PV$^+$ puncta were localized at proximal branches and soma (yellow arrows), whereas VGAT$^+$/PV$^-$ puncta were mainly localized at distal branches (white arrowheads). Only soma and two main branches of the NG2 cell are shown. Inset illustrates a VGAT$^+$/PV$^+$ puncta. (**D**) Percentage of PV$^+$ and PV$^-$ puncta in somata and branches of NG2 cells (n = 4 cells; 6–8 branches per cell). (**E**) Cumulative distributions of PV$^+$ and PV$^-$ puncta in respect to their distance from the soma. Note the restricted distribution for PV$^+$ puncta. **p < 0.01.

The following figure supplements are available for figure 2:

**Figure supplement 1**. Properties of unitary synaptic interneuron-NG2 cell connections.

**Figure supplement 2**. Homogeneous distribution of VGAT$^+$/PV$^+$ puncta around NG2 cells.

subcellular locations with FSI-NG2 cell conductances at proximal contact sites being properly clamped during whole-cell recordings, and NFSI-NG2 cell conductances at more distal contacts being electrotonically filtered; (2) the subunit composition of postsynaptic GABA$_A$Rs is different according to the presynaptic identity of the interneuron (*Pearce, 1993*; *Ali and Thomson, 2008*). It is noteworthy that no correlation was observed for rise and decay times with respect to the postnatal day for either FSI-NG2 cell or NFSI-NG2 cell connections (p > 0.05).

To test for the first possibility, we analyzed the distribution of FSI- and NFSI-GABAergic synaptic contacts on NG2 cells by using 3D confocal reconstructions of the vesicular GABA transporter (VGAT) and PV immunolabeling on biocytin-loaded NG2 cells at PN10, that is, at the peak of connectivity

(*Figure 2C–E—figure supplement 2A*). We considered that VGAT$^+$/PV$^+$ and VGAT$^+$/PV$^-$ puncta corresponded to FSI and NFSI contacts, respectively. VGAT$^+$/PV$^+$ puncta were more abundant than VGAT$^+$/PV$^-$ on the soma, whereas their proportion was relatively similar in NG2 cell branches (*Figure 2C–D*). Furthermore, the distribution of puncta was significantly closer to the soma for VGAT$^+$/PV$^+$ than for VGAT$^+$/PV$^-$ puncta (*Figure 2E*) with a mean distance in branches of 12.8 $\pm$ 1.2 μm and 22.5 $\pm$ 1.3 μm, respectively (p < 0.0001 excluding puncta on the soma). To rule out that the specific segregation of inputs was caused by their uneven distribution around recorded NG2 cells rather than a specific subcellular targeting, we analyzed the distribution of puncta surrounding these cells (*Figure 2—figure supplement 2B*; see 'Materials and methods'). VGAT$^+$/PV$^+$ puncta were homogeneously distributed in the space at the vicinity of analyzed NG2 cells (*Figure 2—figure supplement 2B–C*). These results indicate that FSIs contact preferentially NG2 cell somata and proximal branches, whereas NFSIs mainly contact distal branches. This is consistent with the faster current kinetics observed for unitary FSI-NG2 cell connections and demonstrates a differential distribution of presynaptic inputs on NG2 cells, according to the interneuron identity.

In interneuron–neuron connections, the subunit composition of postsynaptic GABA$_A$Rs can change according to both the location of the receptors in the somato-dendritic compartment and the identity of the presynaptic neuron (*Huang et al., 2007*; *Ali and Thomson, 2008*). Differences on postsynaptic GABA$_A$R subunit composition in NG2 cells may also account for input specificity. We recently showed that GABA$_A$Rs of NG2 cells in the second PN week had a variable and complex subunit composition with around 40% of cortical NG2 cells expressing mRNAs for the GABA$_A$Rs containing γ2 subunits (*Balia et al., 2015*). In agreement with transcript expression, the effect of the positive modulator diazepam (DZP; a benzodiazepine acting on receptors containing the γ2 subunit) on extracellularly evoked currents is also very variable (*Passlick et al., 2013*; *Balia et al., 2015*). To confirm the presence of γ2-subunit protein at postsynaptic sites of NG2 cells, we performed triple immunostainings against γ2, VGAT (presynaptic marker), and NG2 (a marker of NG2 cell membranes) in NG2-DsRed mice at PN10 (*Figure 3A*). We observed numerous VGAT$^+$/γ2$^+$ and VGAT$^+$/γ2$^-$ puncta on NG2$^+$ cell membranes of the soma and branches as previously observed for PV/VGAT$^+$ in biocytin-loaded cells. In addition, VGAT$^+$/γ2$^+$ synaptic puncta on neurons can be clearly distinguished from those present on NG2 cells at this developmental stage (*Figure 3A*), corroborating the presence of VGAT$^+$ puncta on these progenitors.

To evaluate whether the presynaptic identity is associated to the presence of γ2 subunits in GABA$_A$Rs at postsynaptic sites, we bath-applied DZP (10 μM) during paired recordings of mice from PN9 to PN13. This benzodiazepine significantly increased the amplitude of unitary synaptic currents in connected FSI-NG2 cell pairs without modifying the paired-pulse ratio (PPR), whereas it did not affect currents of unitary NFSI-NG2 cell connections (*Figure 3B–C*; PPR of 0.55 $\pm$ 0.08 and 0.38 $\pm$ 0.05 with and without DZP for FSIs and 0.46 $\pm$ 0.13 and 0.54 $\pm$ 0.14 with and without DZP for NFSIs, respectively, p > 0.05). Hence, FSIs preferentially targeted proximal postsynaptic sites of NG2 cells containing GABA$_A$Rs with γ2 subunits, whereas NFSIs favored more distal sites lacking this subunit.

Overall, the analyses of current kinetics, 3D reconstructions of PV/VGAT$^+$ puncta, and pharmacology of γ2-containing GABA$_A$Rs demonstrated that FSIs and NFSIs target distinct subcellular segregated domains of NG2 cells, containing different GABA$_A$Rs. A highly specific spatial arrangement of synaptic inputs, thus, exists at the subcellular level in NG2 cells.

## Restricted number of release sites per interneuron

The extracellular stimulation of neuronal fibers induces postsynaptic currents in NG2 cells that can reach hundreds of pA in the somatosensory cortex (*Maldonado et al., 2013*). However, current amplitudes of unitary interneuron-NG2 cell connections are very small compared to extracellularly evoked currents, suggesting that a single NG2 cell is densely connected by many interneurons in the network, but that a single interneuron contacts a glial progenitor through a restricted number of release sites. To determine whether interneuron-NG2 cell connections consisted of single- or multiple-release sites, we examined the response probabilities by using paired-pulse stimulations of presynaptic neurons (*Angulo et al., 1999*). In 8 out of 18 connections, quantal analysis revealed no statistical differences between cumulative distributions of PSC1 and PSC2 amplitudes, excluding failures, although the response probabilities were significantly higher for PSC1 (*Figure 4A,B,E,F*; see 'Materials and methods'). This implies that PSCs result from the release of only one quantum of transmitters, and therefore, a single vesicle was released per connection (*Stevens and Wang, 1995*).

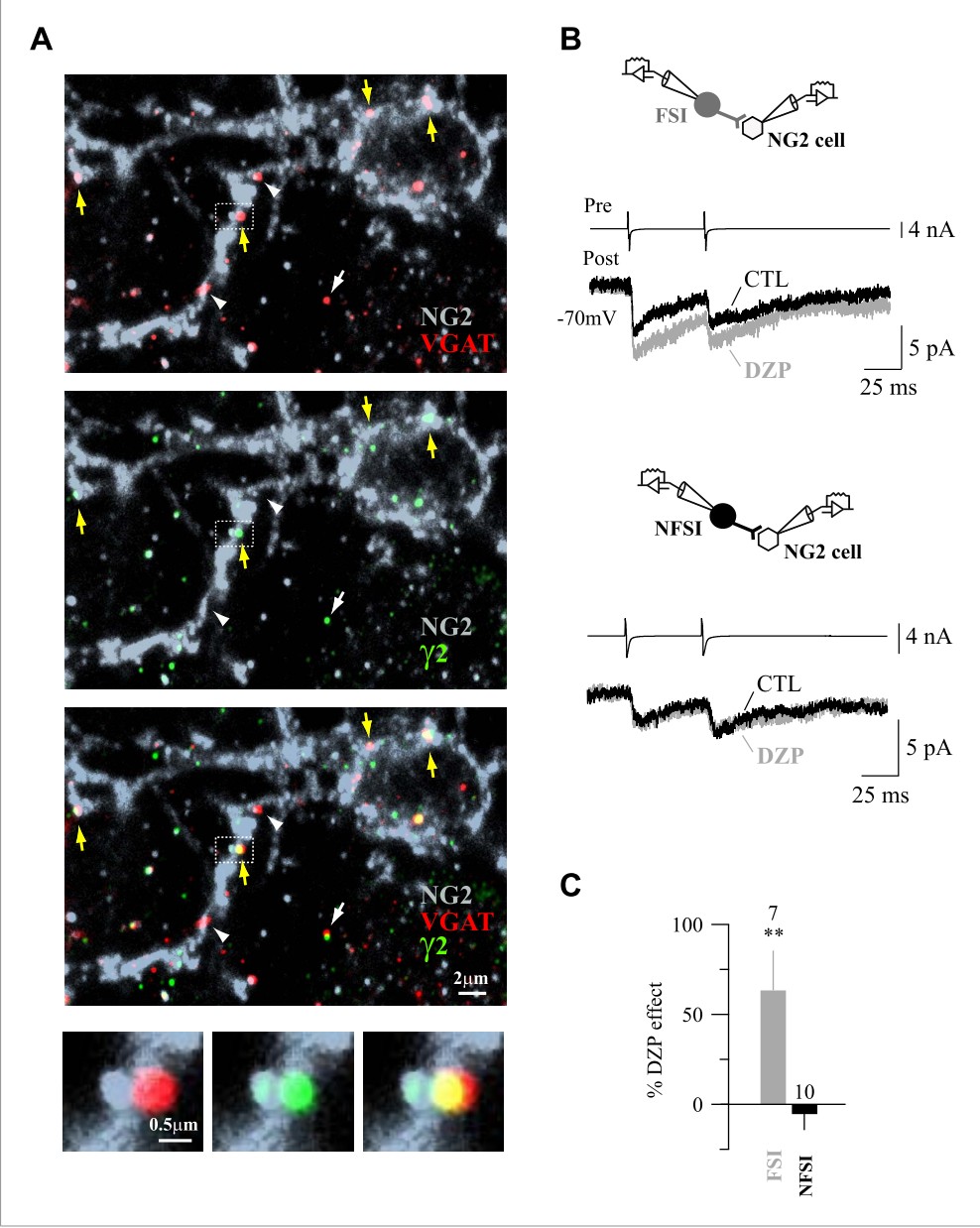

**Figure 3**. Expression of γ2 subunit of GABA$_A$Rs at FSI-NG2 cell synapses. (**A**) Confocal images of VGAT$^+$ (red) and γ2$^+$ (green) puncta on NG2$^+$ membranes (gray) of a NG2 cell at PN10 (objective 63x; stack of 8 Z sections, each 0.32 μm). As expected from previous data (**Balia et al., 2015**), numerous VGAT$^+$/γ2$^+$ (yellow arrows) and VGAT$^+$/γ2$^-$ (white arrowheads) puncta on NG2$^+$ soma and branches were observed. Note that neuronal VGAT$^+$/γ2$^+$ puncta are clearly distinguished from those present on NG2 cells at this developmental stage (white arrow). (**B**) DZP effect on PSC amplitudes in two NG2 cells connected, respectively, to a FSI (top) and a NFSI (bottom). (**C**) Histogram comparing DZP effect on PSCs evoked by FSIs and NFSIs. The number of tested cells is indicated on top of histogram bars. **p < 0.01.

The mean amplitude of PSCs without failures corresponded to a mean quantal size of −7.71 ± 0.71 pA in our recording conditions. These interneurons, thus, innervate NG2 cells probably through one release site, independent of the presynaptic interneuron identity and the postnatal day (5 FSIs and 3 NFSIs displayed single vesicular release at postnatal days from PN8 to PN13).

For the other 10 pairs, quantal analysis showed statistical differences between cumulative amplitude distributions and response probabilities of PSC1 and PSC2, indicating that more than one

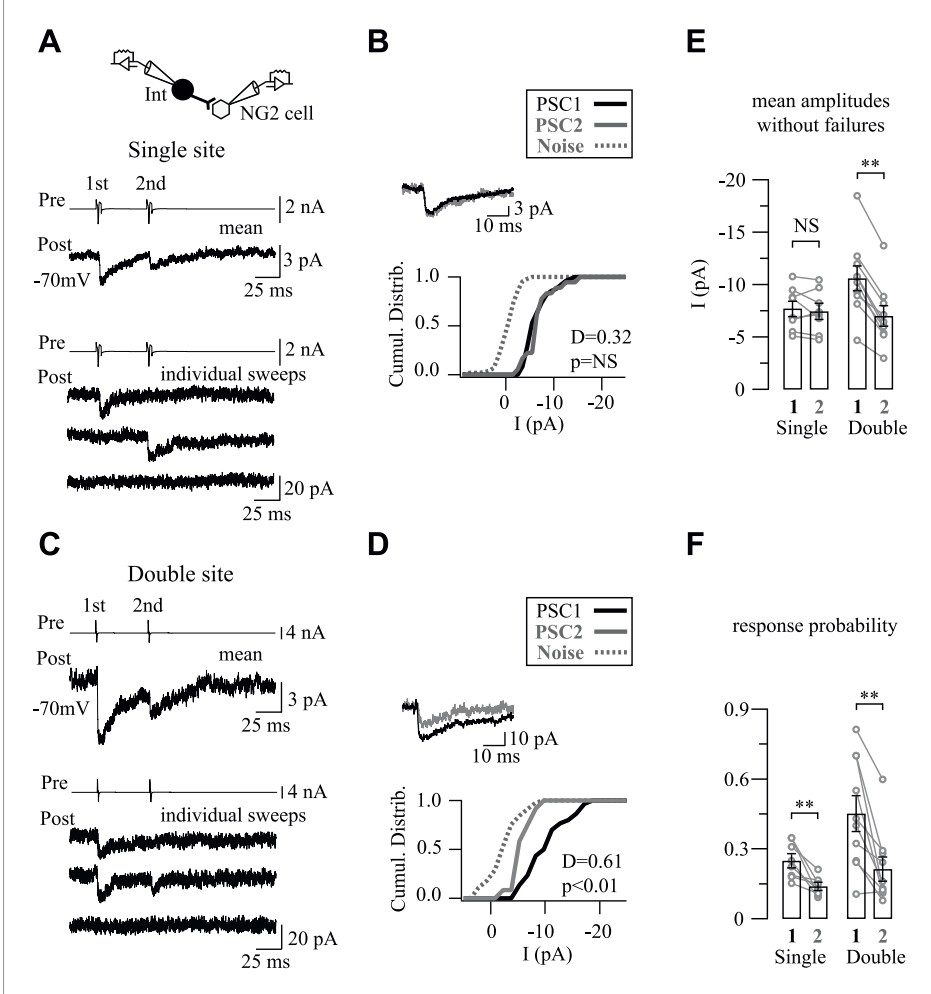

**Figure 4**. One or two release sites per interneuron-NG2 cell connection. (**A**, **C**) Unitary connections with single (**A**) or double (**C**) vesicular release. Mean (middle traces) and individual (bottom traces) PSCs recorded in NG2 cells in response to two action currents evoked in presynaptic interneurons (top traces). (**B**, **D**) Cumulative distributions of PSC1 and PSC2 without failures for connections shown in **A** and **C**. Insets show superimposed mean PSC1 and PSC2 without failures. (**E**, **F**) Histograms of amplitudes without failures (**E**) and response probabilities (**F**) of PSC1 and PSC2 for connections with single and double vesicular release.

vesicle were released per connection (*Figure 4C–F*). Interestingly, amplitudes and probabilities of response were around twice for PSC1 and the same for PSC2 when compared to those of PSC1 resulting from a single vesicular release (*Figure 4E–F*; *Figure 2—figure supplement 1F–J*). Thereby, these connections exhibited double vesicular release and probably displayed double release sites.

In addition to the specific spatial distribution of GABAergic inputs impinging on NG2 cells, interneurons established a point-to-point communication with these progenitors through single- or double-release sites. This restricted innervation contrasts with the ability of interneurons to innervate other neurons through multiple release sites at the same PN stage (*Pangratz-Fuehrer and Hestrin, 2011*).

## Local spatial arrangement of interneuron-NG2 cell connections

The highly organized interneuron-NG2 cell connectivity at the subcellular level suggests that NG2 cells are specifically wired by interneurons inside functional neural assemblies, probably forming a specific network. To test this hypothesis, we investigated the arrangement of functional interneuron-NG2 cell microcircuits in layer V from PN8 and PN11 and compared it with that of interneuron–pyramidal cell microcircuits.

To establish GABAergic connectivity maps of NG2 cells and pyramidal neurons, we exploited the flexibility and high-spatial precision of holographic photolysis (*Lutz et al., 2008*; *Zahid et al., 2010*) (*Figure 5—figure supplement 1A–B*). By generating precise light patterning in real time, this optical method enables the photolysis of caged neurotransmitters such as MNI-glutamate to photostimulate neurons (*Lutz et al., 2008*; *Zahid et al., 2010*). This cage compound is one of the most efficient in terms of release of glutamate by light and of stability at physiological pH and temperature (*Matsuzaki et al., 2001*). However, as many other cages, it has a blocking effect on GABA$_A$Rs in brain slices (*Fino et al., 2009*) (*Figure 5—figure supplement 1C*). For this reason, we first searched for an appropriate concentration of MNI-glutamate for which the effect on GABA$_A$Rs of NG2 cells is minimal while triggering efficiently action potentials in interneurons at a single-cell resolution. We found that a concentration of 50 μM MNI-glutamate fulfilled these two prerequisites (*Figure 5—figure supplement 1C,D* and *Figure 5—figure supplement 2*).

To build GABAergic connectivity maps of recorded cells, interneurons were sequentially photostimulated in the excitation field using a 5-μm light spots, while patched cells were recorded with CsMeS-based intracellular solution at 0 mV, the reversal potential of their ionotropic glutamatergic receptors (*Figure 5A–D*). Outward GABA$_A$R-mediated PSCs, sensitive to SR95531, were induced in recorded cells by photostimulation of nearby interneurons (*Figure 5E*). We considered as connected pairs those showing both PSCs detected in averaged traces with a threshold of 2 times the standard deviation of the noise and an increased occurrence probability of individual PSCs within 100 ms after photostimulation when visualized in raster plots (*Figure 5A–D*). This time window corresponds to the latency after photostimulation required to trigger an action potential in interneurons (*Figure 5—figure supplement 3A–B*). Indeed, action potential generation with light through the activation of glutamate receptors cannot be precisely controlled as with patch-clamp recordings. In most cases, action potential generation of interneurons was delayed with respect to the photostimulation time and displayed a spike jitter (*Figure 5—figure supplement 3A–B*). As a consequence, there is also a variable latency and jitter in photo-evoked PSCs during mapping experiments (*Figure 5B,D*, insets). Nevertheless, to confirm irrefutably the monosynaptic nature of the connections, the photo-activated interneuron inducing PSCs in a postsynaptic cell was patched with a second pipette to test its connectivity with paired recordings (*Figure 5—figure supplement 3D,E*). Three out of three-tested pairs were truly connected with the postsynaptic cell.

Although most interneurons were photo-activated by less than 3-ms pulses, some others required a pulse duration up to 7 ms to reach their action potential threshold (*Figure 5—figure supplement 3C*). In order to take into account different action potential thresholds, we systematically increased or decreased the duration of the laser pulse for each interneuron. Furthermore, to ensure the spatial selectivity of the system, we displaced the spot away from the soma, resulting in the disappearance of the response (*Figure 5A–D—figure supplement 3D*). This allows us to reveal false positive and negative connections, respectively (*Figure 5A,C,F*). Only in rare cases, the unambiguous discrimination of connected interneurons was not possible, and thus, these targeted interneurons were not considered in cell maps (2.8% of all tested interneurons). The online evaluation of each photostimulated interneuron allowed us to set reliable maps of connectivity for recorded cells.

As expected for connections with a limited number of release sites, the amplitudes of PSCs induced in NG2 cells by photostimulation were smaller than those in pyramidal neurons (*Figures 5A, C, 6B–E*). Either in the sample of tested cells (*Figure 5G*) or in maps showing at least one connection (*Figure 5H*), the probability to find a connected pair was lower for NG2 cells than for pyramidal neurons. Interestingly, connectivity maps of NG2 cells involved very local microcircuits with connections at interneuron-NG2 cell intersomatic distances never exceeding 70 μm and cumulative distributions of intersomatic distances significantly different for connected and unconnected interneurons (*Figure 6G*). Moreover, the connection probability highly decreased after intersomatic distances of 50 μm (*Figure 6G*). On the contrary, connectivity maps of pyramidal cells were relatively homogeneous within 100 μm with similar cumulative distributions of intersomatic distances between connected and unconnected cells in agreement with previous reports (*Fino and Yuste, 2011*) (*Figure 6H*). The local architecture of interneuron-NG2 cell connections, thus, did not result from the impossibility to detect connections over 70 μm with holographic photolysis. These findings reveal that interneuron-NG2 cell microcircuits are arranged according to a specific connectivity pattern that follows a very local microarchitecture.

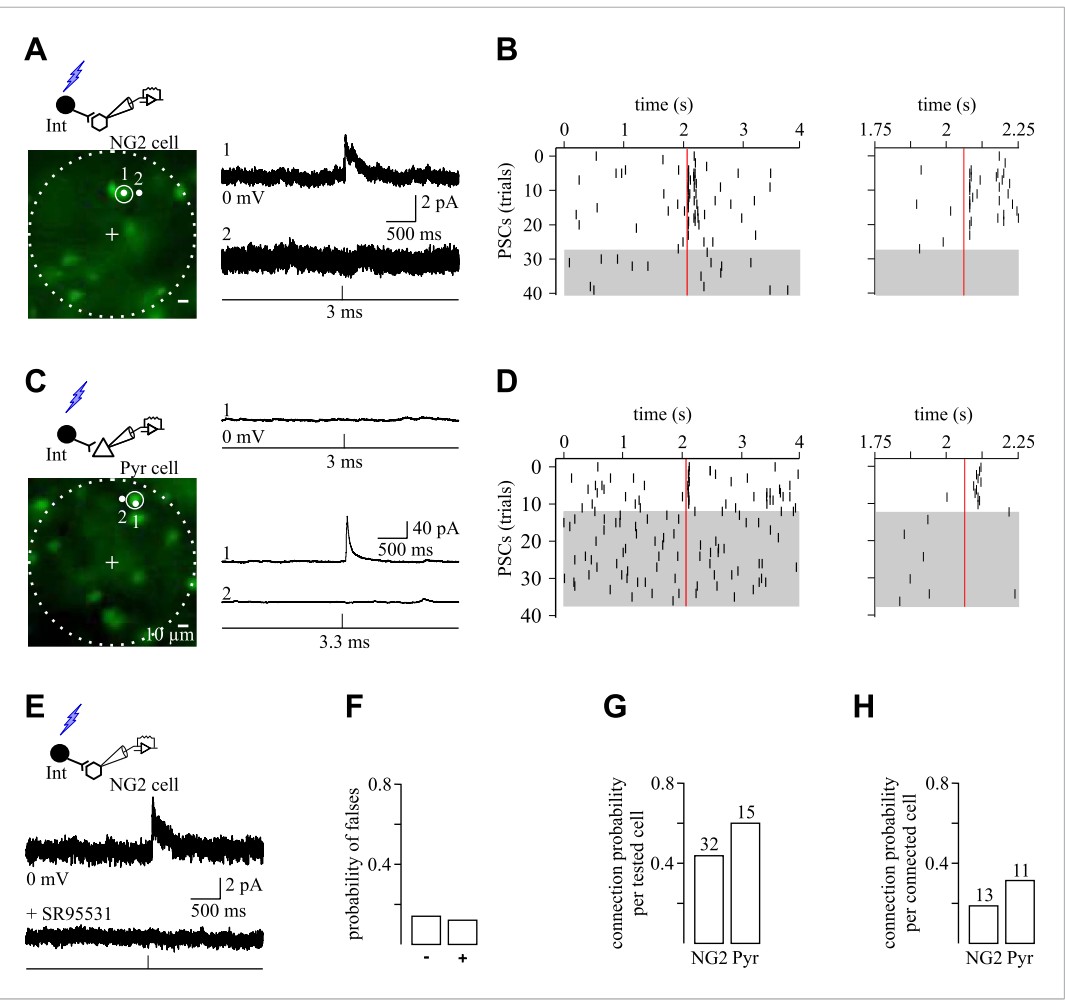

**Figure 5**. Spatial selectivity of holographic photostimulation to detect unitary interneuron-NG2 cell connections. (**A**, **C**) Excitation fields (dashed circles) in epifluorescent images of Venus+ interneurons. Recorded NG2 cell (**A**) and pyramidal neuron (**C**) are in the center (+, non-visible). A 3-ms photostimulation of an interneuron (**A**; spot 1) induces unitary PSCs in a NG2 cell held at 0 mV and recorded with a CsMeS-based intracellular solution (top trace; average of 28 traces). The spatial selectivity of this connection is confirmed by displacing the illumination spot near the targeted soma (spot 2; bottom trace; average of 12 traces). (**B**) Raster plot of GABAergic synaptic events from the recorded NG2 cell illustrated in **A**. Each tick corresponds to a PSC. Note that random and sparse spontaneous synaptic currents are observed 2 s before and after interneuron photostimulation (red vertical line), whereas photo-evoked synaptic events reproducibly occur within 100 ms after the photostimulation. Photo-evoked events disappear when the 5-μm spot is moved to spot 2 (gray box). Note that in some single traces more than one postsynaptic event was elicited upon photo-stimulation (insets right) and that the average response in **A** displays two peaks. In these examples, the targeted presynaptic interneuron probably elicited more than one action potential. (**C**) A 3-ms photostimulation of an interneuron (**C**; spot 1) does not induce unitary synaptic currents in a pyramidal neuron held at 0 mV and recorded with a CsMeS-based intracellular solution (top trace). The excitation time was increased to test for a possible false negative connection. An increase in the excitation time of the interneuron to 3.3 ms induces unitary PSCs in the pyramidal neuron (spot 1; middle trace; average of 11 traces) that disappear when the spot is displaced 5 μm apart, confirming the photostimulation selectivity (spot 2; bottom trace; average of 26 traces). (**D**) Raster plot of GABAergic synaptic events from the recorded pyramidal neuron illustrated in **C**. Note that random spontaneous synaptic currents are observed 2 s before and after interneuron photostimulation (3.3 ms; red vertical line), whereas synaptic events reproducibly occur within 100 ms after photostimulation (inset right). These events disappear when the 5-μm spot is moved to position 2 (gray box). Failures of response were rarely observed in pyramidal neurons. (**E**) Averaged unitary PSC photo-induced in a recorded NG2 cell (middle trace; average of 9 traces) and completely abolished by 5 μM SR95531 (bottom trace; average of 13 traces). (**F**) Probability of encountering false negative and false positive connections. Unspecific connections were discriminated by
*Figure 5. continued on next page*

*Figure 5. Continued*

changing the pulse duration of the laser and the position of the spot as in **A** and **C**. (**G**, **H**) Connection probabilities for all tested cells (**G**) and for cells showing at least one connection (**H**).

The following figure supplements are available for figure 5:

**Figure supplement 1**. Optical set-up and effect of MNI-glutamate on GABA$_A$ receptor-mediated currents in NG2 cells.

**Figure supplement 2**. Spatial selectivity of holographic photostimulation of targeted Venus[+] interneurons.

**Figure supplement 3**. Monosynaptic connections with holographic photolysis.

## Transient connectivity coincides with a switch to differentiation

Our results demonstrate that GABAergic innervation of NG2 cells is spatially organized following a specific arrangement of inputs at the subcellular and network levels during the second postnatal week, that is, a critical period for NG2 cell differentiation. Indeed, the production of premyelinating oligodendrocyte in the cerebral cortex starts during the first postnatal week and reaches a peak at PN14 when compared to PN21 (*Trapp et al., 1997*; *Baracskay et al., 2002*). We investigated, therefore, whether GABAergic innervation of NG2 cells during this period occurs in conjunction with the active phase of differentiation of these progenitors. For these, we analyzed the connection probability of paired recordings on a daily basis, from PN8 to PN13, and compared it with the NG2 cell differentiation process in layer V. We observed that the connection probability reached a peak at PN10, when we found 44% of connected pairs, and then decreased (*Figure 7A*), confirming the transient GABAergic innervation of NG2 cells (*Vélez-Fort et al., 2010*). This time course of the connectivity is specific for NG2 cells since the probability of interneuron-to-neuron connection is known to increase with postnatal cortical development (*Pangratz-Fuehrer and Hestrin, 2011*; *Yang et al., 2012*).

To establish whether the high degree of synaptic connectivity at PN10 correlates with the cortical NG2 cell-differentiation process, we performed immunostainings against CC1, a specific marker of differentiated oligodendrocytes, and Olig2, a specific marker of the oligodendrocyte lineage, in NG2-DsRed mice from PN9 to PN13 (*Figure 7B–F*; see 'Materials and methods'). The density of CC1[+]/Olig2[+] cells was low at PN9 and PN10 and significantly increased from PN11 (*Figure 7A–F*). Therefore, the peak of synaptic connectivity of cortical NG2 cells at PN10 coincides with a switch to a massive NG2 cell differentiation occurring between PN10 and PN11 in cortical layer V (*Figure 7A*). To further analyze whether the transient GABAergic connectivity of NG2 cells and the switch to oligodendrocyte differentiation were related processes, we tested whether the Na$^+$ current density of NG2 cells correlates with the frequency of spontaneous synaptic events at different postnatal days. Indeed, it has been shown that both the amplitude of Na$^+$ currents and the synaptic current frequency decrease when these progenitors undergo differentiation (*De Biase et al., 2010*; *Kukley et al., 2010*). As expected from paired recordings and previous studies (*Vélez-Fort et al., 2010*; *Balia et al., 2015*), the frequency of spontaneous synaptic activity increases from PN8 to PN10 and then decreases at PN13 ($0.12 \pm 0.02$ Hz, $0.24 \pm 0.05$ Hz, $0.12 \pm 0.02$ Hz, respectively; $p < 0.05$ for PN10). However, no statistical differences were observed either on the averaged amplitudes of spontaneous synaptic currents or Na$^+$ current densities between PN8, PN10, and PN13 (amplitude of synaptic currents: $-10.45 \pm 1.04$ pA, $-12.75 \pm 1.27$ pA, $-12.07 \pm 0.90$ pA, respectively; Na$^+$ current densities: $22.59 \pm 2.99$ pA/pF, $23.46 \pm 3.56$ pA/pF, $26.10 \pm 3.72$ pA/pF, respectively; $p > 0.05$). Interestingly, while no correlation was observed between the Na$^+$ current density and the frequency of spontaneous synaptic events at PN8 and PN13 when connectivity is low, a positive correlation was observed at PN10 when connectivity is high and the switch to differentiation starts (*Figure 7—figure supplement 1*). These data suggest that interneurons build up a large number of functional synapses in more immature NG2 cells at the onset of massive oligodendrocyte differentiation (PN10) and then, GABAergic synapses disassemble. The transient NG2 cell-synaptic connectivity is thus not only spatially, but also temporally regulated in coordination with the active differentiation phase of these progenitors in the second PN week.

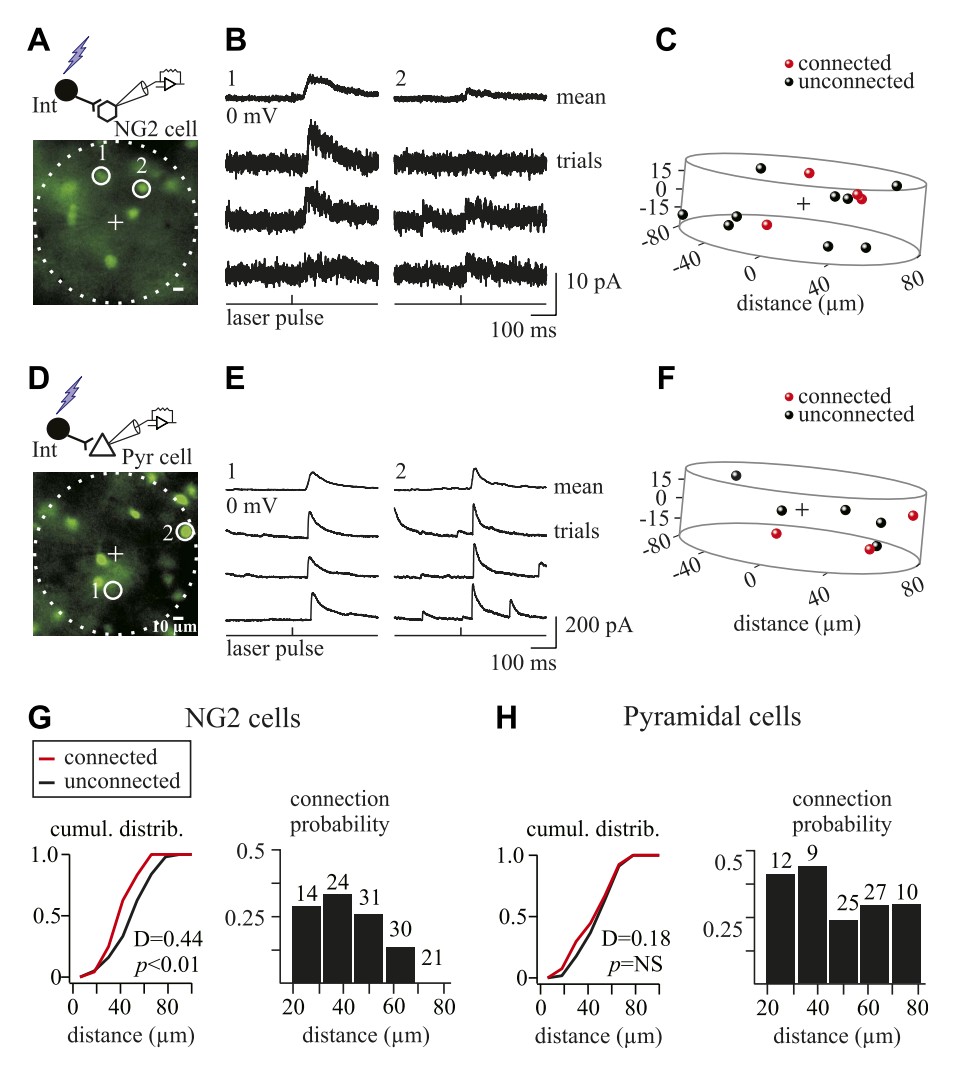

**Figure 6**. GABAergic connectivity maps of NG2 cells and pyramidal neurons. (**A**, **D**) Excitation fields (dashed circles) in epifluorescent images of Venus[+] interneurons. Recorded NG2 cell (**A**) and pyramidal neuron (**D**) are in the center (+, non-visible). (**B**, **E**) Photostimulation of interneurons (1 and 2) in **A** and **D** induces unitary PSCs in a recorded NG2 cell (**B**) and pyramidal neuron (**E**). (**C**, **F**) Connectivity maps within $1.05 \times 10^6$ $\mu m^3$ volume of cells in **A** and **D** showing connected (red) and unconnected (black) interneurons. (**G**, **H**) Cumulative distribution of connected and unconnected interneurons (left) and distribution of connection probabilities in respect to intersomatic distances (right) between interneurons and either NG2 cells (n = 13) or pyramidal neurons (n = 11) displaying at least one connection.

## Discussion

Our findings uncover both spatially and temporally structured interneuron-NG2 cell connections, from the subcellular level to the cortical network during a critical period of NG2 cell differentiation. We demonstrate that NG2 cells form a transient and highly organized network with interneurons that is characterized by a high-connection probability at PN10, a specific distribution of synaptic inputs on cell bodies and branches, a restricted number of contacts per interneuron, and very local connectivity maps. GABAergic innervation of NG2 cells, thus, appears as a finely regulated process that follows its own logic, which cannot be inferred from previous studies on NG2 cells and on classical neuronal synapses.

In the neocortex, different classes of interneurons, guided by genetically determined mechanisms, are known to target distinct subcellular domains of pyramidal neurons, allowing a differential compartmentalized signal processing (*Huang et al., 2007*). We demonstrate that input-specific

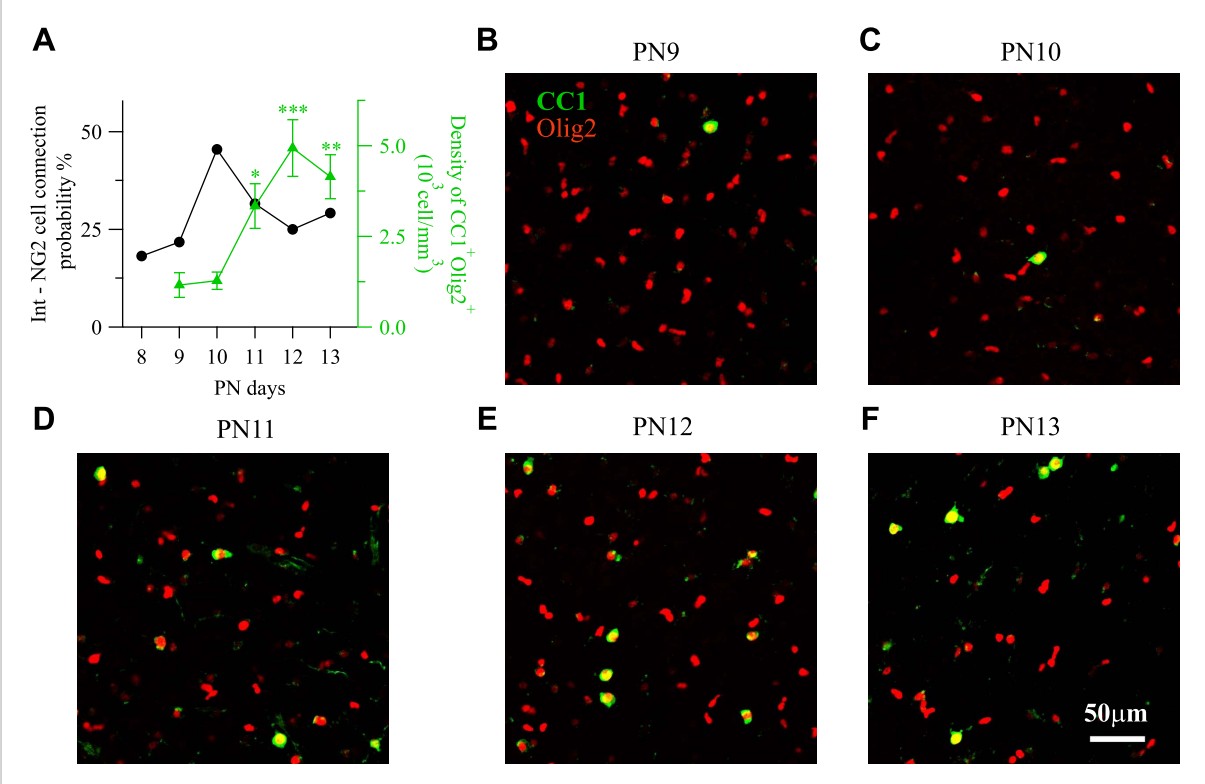

**Figure 7**. Time course of interneuron-NG2 cell connectivity and NG2 cell differentiation during the second PN week. (**A**) Connection probability for interneuron-NG2 cell pairs and $CC1^+/Olig2^+$ cell density as a function of postnatal days (19–25 pairs tested per day). Note that the peak of NG2 cell connection probability at PN10 precedes a large increase in $CC1^+/Olig2^+$ differentiated oligodendrocytes at PN11. *p < 0.05; **p < 0.01, ***p < 0.001. (**B**–**F**) $CC1^+/Olig2^+$ and $CC1^-/Olig2^+$ cells in layer V at different PN days (stacks of 10 Z-sections; each 1 μm).

The following figure supplement is available for figure 7:

**Figure supplement 1**. $Na^+$ current density and frequency of spontaneous activity of NG2 cells correlates at PN10, but not at PN8 and PN13.

projections also exist for interneuron-NG2 cell connections. NG2 cells compartmentalize FSI- and NFSI-input regions as revealed by distinct anatomical locations of $PV^+$ and $PV^-$ contacts and by a different distribution of $GABA_A$Rs with or without $\gamma_2$ subunits at postsynaptic sites. FSIs and NFSIs impinging on these glial progenitors, thus, probably encode distinct information that regulates different cellular NG2 cell processes. Located at proximal sites and somata, FSI synapses could regulate NG2 cell differentiation by controlling gene expression, whereas distal NFSI synapses could affect motility or migration. Indeed, it has been suggested that GABA probably promotes NG2 cell migration through the activation of $GABA_A$ receptors (*Tong et al., 2009*).

NG2 cells receive inputs from both FSIs and NFSIs, but FSIs are proportionally more connected to these progenitors, suggesting that they play a major role in controlling NG2 cell activity during early stages of postnatal development. Compared to adult animals, immature FSIs possess different electrophysiological properties, such as high-input resistance, relatively low-membrane time constants, and a certain degree of spike-frequency adaptation, and are thus unlikely to perform the same function that in mature circuits (*Okaty et al., 2009*). In fact, FSIs probably do not contribute to coordinating cortical neuronal activity in the perinatal period since FSI-pyramidal cell connections are detected only from PN5 (*Pangratz-Fuehrer and Hestrin, 2011*). FSIs start to influence cortical activity only during the second postnatal week (*Pangratz-Fuehrer and Hestrin, 2011*), and as shown here, it is also at this period that they constitute a significant, transient presynaptic input to NG2 cells. Synaptic communication between FSIs and NG2 cells could, therefore, play a role in the maturation of FSIs in the developing neuronal network.

Although local cortical interneurons are believed to lack myelin because they possess relatively short axons that do not project outside the cortex, *Somogyi et al. (1983)* demonstrated the presence of myelin enwrapping the axon of basket cells, most probably FSIs, in cats (*Somogyi et al., 1983*). Since myelination is an activity-dependent process (*Zalc and Fields, 2000*), an interesting possibility would be that the restricted point-to-point GABA release at unitary interneuron-NG2 cell connections encodes a signal that triggers NG2 cell differentiation and myelination of the presynaptic interneuronal axon, preferentially that of FSIs. Myelination of neocortical FSIs is likely to occur since myelin is important for the organization of Kv1 channels (*Rasband et al., 1998*), and these channels have been proposed to mediate the variability in FSI-firing patterns (*Golomb et al., 2007*). In addition, axonal signals are essential to regulate oligodendrocyte production and survival (*Barres et al., 1992*; *Trapp et al., 1997*), and interneuronal axons transiently contact NG2 cells during active-cortical oligodendrogenesis.

Many one-photon uncaging systems have been used to study the synaptic connectivity patterns in brain regions, but their low-spatial resolution does not allow for photo-stimulating neurons at a single-cell level (*Dantzker and Callaway, 2000*; *Shepherd et al., 2003*; *Yoshimura et al., 2005*). On the contrary, we show that one-photon holographic photolysis is a suitable alternative tool to elicit action potentials in neurons at single-cell resolution and find unitary connections. Using this technique, we demonstrate that interneuron-NG2 cell connections display a very local arrangement in the network. This difference cannot be explained exclusively by the small volume occupied by NG2 cells. Indeed, axons of interneurons that ramify extensively and travel long did not innervate NG2 cells over 70 μm. In fact, the connectivity of NG2 cells already decreases at interneuron-NG2 cell intersomatic distances of 50 μm (*Figure 6G*). It has been reported that no specificity exists between the spatial profiles of interneuron-pyramidal cell connectivity maps, and thus, that pyramidal cells do not form specific networks in the neocortex (*Fino and Yuste, 2011*; *Packer et al., 2014*). In contrast, interneuron-NG2 cell connectivity maps follow a local spatial arrangement, reflecting a focal control of NG2 cell activity by interneurons. This microarchitecture supposes a close relationship between the proximal part of interneuronal axons and NG2 cells. The existence of a local network formed by interneurons and NG2 cells, embedded within the developing neuronal network, implies the involvement of selective molecular and cellular mechanisms ensuring this local connectivity. These mechanisms remain to be elucidated.

In the developing postnatal brain, the proper maturation of interneuron–neuron microcircuits requires the interactions between intrinsic genetic programs and neuronal activity (*Cossart, 2011*). We demonstrate that the emergence of properly organized GABAergic neuronal microcircuits is not only confined to neurons, but also includes a non-neuronal cell type. Interestingly, after the peak of synaptic connectivity at PN10, the decrease of GABAergic synaptic innervation of NG2 cells is accompanied by other relevant physiological changes in these progenitors: (1) there is a decrease in the amplitude of $GABA_AR$-mediated miniature events (*Balia et al., 2015*); (2) Kir channels start to be upregulated (*Kressin et al., 1995*); and (3) the first wave of oligodendrocytes arising from Nkx2.1-expressing precursors of MGE and the anterior entopeduncular area is eliminated (*Kessaris et al., 2006*). An intriguing question for future research is whether cortical NG2 cell development onto oligodendrocytes occurs under the control of interneuronal activity in the developing neuronal network. The GABAergic synaptic input probably does not control per se cortical oligodendrocyte production, which is a protracted process that occurs during several weeks, even after the loss of functional synapses. However, the coincidence between the peak of connectivity (PN10) and the switch to a massive NG2 cell differentiation from PN10 raises the possibility that the interneuron-NG2 cell network sets precisely the onset of oligodendrogenesis occurring in deep layers of the neocortex during the second postnatal week. This would explain why a transient and structured NG2 cell connectivity is necessary.

## Materials and methods

### Acute slice preparation and electrophysiology

All experiments followed European Union and institutional guidelines for the care and use of laboratory animals. Acute parasagittal slices (300 μm) of the barrel cortex with an angle of 10° to the sagittal plane were obtained from a double VGAT-Venus;NG2-DsRed transgenic mouse (*Ziskin et al., 2007*; *Wang et al., 2009*), as previously described (*Vélez-Fort et al., 2010*). Excitation light to visualize Venus and DsRed fluorescent proteins was provided by Optoled Light Sources (Blue and Green Optoleds; Cairn Research, UK), and images were collected with an iXon+ 14-bit digital camera

(Andor Technology, UK) through an Olympus BX51 microscope equipped with a 40× fluorescent water-immersion objective. Excitation and emission wavelengths were obtained by using, respectively, 470- and 525-nm filters for Venus and 560- and 620-nm filters for DsRed. The Imaging Workbench 6.0 software (Indec Biosystems, USA) was utilized to acquire and store images for off-line analysis. Patch-clamp recordings were performed at RT or 33°C using an extracellular solution containing (in mM): 126 NaCl, 2.5 KCl, 1.25 $NaH_2PO_4$, 26 $NaHCO_3$, 20 glucose, 5 pyruvate, 3 $CaCl_2$, and 1 $MgCl_2$ (95% $O_2$, 5% $CO_2$). NG2 cells were recorded with different intracellular solutions according to the experiment and containing (in mM): either 130 CsCl or 125 $CsCH_3SO_3H$ (CsMeS), 5 4-aminopyridine, 10 tetraethylammonium chloride, 0.2 EGTA, 0.5 $CaCl_2$, 2 $MgCl_2$, 10 HEPES, 2 $Na_2$-ATP, 0.2 Na-GTP, and 10 $Na_2$-phosphocreatine (pH ≈ 7.3). Presynaptic interneurons were recorded with an intracellular solution containing (in mM): 130 K-gluconate (KGlu), 10 GABA, 0.1 EGTA, 0.5 $CaCl_2$, 2 $MgCl_2$, 10 HEPES, 2 $Na_2$-ATP, 0.2 Na-GTP, and 10 $Na_2$-phosphocreatine (pH ≈ 7.3). Potentials were corrected for a junction potential of −10 mV when using CsMeS and KGlu-based intracellular solutions. Whole-cell recordings were obtained using Multiclamp 700B, filtered at 4 kHz, and digitized at 20 kHz. Digitized data were analyzed off-line using pClamp10.1 software (Molecular Devices), Neuromatic package (http://www.neuromatic.thinkrandom.com/) and Spacan (http://www.spacan.net) within IGOR Pro 6.0 environment (Wavemetrics, USA). $Na^+$ current densities and frequency of spontaneous synaptic activity were analyzed as we previously described (*Vélez-Fort et al., 2010*; *Balia et al., 2015*).

## Paired recordings

Paired recordings were performed between a Venus[+] interneuron and a DsRed[+] NG2 cell both held at −70 mV with two patch pipettes. To test for a functional connection, paired-pulse stimulation was applied to the interneuron in voltage-clamp mode to elicit action currents at 8-s intervals (1 ms, 80 mV pulse; 50 ms paired-pulse interval). This protocol allows for a precise timing of action potential generation in interneurons. We considered as a unitary connection, those pairs showing averaged PSCs in NG2 cells larger than 2 times the standard deviation of the noise. To evaluate the recovery from depression, we applied two test pulses using interstimulus intervals ranging from 10 ms to 250 ms. Paired-pulse ratios were calculated as PSC2/PSC1. Quantal analyses were performed on 18 out of 38 connections for which 100 or more traces were recorded, and individual PSCs in single traces could be differentiated from the noise using a detection threshold of 2 times the standard deviation.

## Holographic photolysis

The holographic setup was adapted to the Olympus microscope as previously described (*Figure 5—figure supplement 1A*) (*Lutz et al., 2008*; *Zahid et al., 2010*). Briefly, a 405-nm diode CW-laser (CUBE 405-100, Coherent) was used for uncaging experiments. The output beam was expanded (6×) to match the input window of a LCOS-SLM (X10468-01, Hamamatsu), which operates in reflection mode. The device was controlled by a custom-designed software described in *Lutz et al. (2008)* that calculated the corresponding phase hologram and addressed the pattern to the LCOS-SLM, given a target intensity distribution at the focal plane of the microscope objective. The SLM plane was imaged at the back aperture of the microscope objective through a telescope (L1, f1 = 350 mm; L2, f2 = 180 mm). The undiffracted component (zero-order spot) was masked at the focal plane of L1 using a coverslip with a black dot.

Acute slices transferred into the recording chamber were perfused with the extracellular solution at 2–3 ml/min using a recycling bubbled system (10 ml) that allows for the continuous perfusion of the caged MNI-glutamate (50 μM). Selective photostimulation of interneurons during patch-clamp recordings of either NG2 cells or pyramidal neurons was obtained with 5-μm illumination spots during 3–8 ms and a laser power of ~12 mW under the objective. Patched cells were recorded with CsMeS-based intracellular solution and held at 0 mV to minimize the direct photo-activation of their glutamatergic receptors in the excitation field. The protocol consisted in using an initial photo-stimulation of 3 ms that photo-evoked single or few action potentials in most targeted interneurons (*Figure 5—figure supplement 3C*). False negative and false positive connections were discerned by changing the laser time pulse and by moving the illumination spot outside the soma of the targeted interneuron (*Figure 5A–D,F*). We considered as photo-induced unitary PSCs those that: (1) showed an increased occurrence probability of individual PSCs within 100 ms after photostimulation when visualized in raster plots (this time window corresponded to the time needed for interneurons to spike

after the photostimulation, *Figure 5—figure supplement 3B*); and (2) were detected in averaged traces with a threshold of 2 times the standard deviation of the noise.

## Immunostainings and puncta analysis

For CC1 and Olig2 immunostainings, NG2-DsRed mice of the same litters were perfused intracardially with phosphate buffer saline (PBS) alone followed by 0.15 M phosphate buffer, pH 7.4 (PB) containing 4% paraformaldehyde at PN9–PN13 (n = 6 litters and 5–7 animals per age). Brains were removed and placed in a 4% paraformaldehyde solution overnight. Then, brain slices (50 µm) were prepared in PBS ice-cold solution (4°C), permeabilized with 0.2% triton X-100 and 4% Normal Goat Serum (NGS) for 1 hr, and incubated one night with antibodies diluted in a 0.2% triton X-100 solution and 2% NGS. For VGAT, γ2, and NG2 triple immunostainings, animals were similarly perfused, and brains placed in 4% paraformaldehyde solution for 1 hr. Then, brain slices (100 µm) were prepared in PBS ice-cold solution (4°C), permeabilized with 1% triton X-100 and 4% NGS for 1 hr, and incubated three nights with antibodies diluted in a 0.2% triton X-100 solution and 2% NGS. Double immunostainings were performed by combining rabbit anti-Olig2 (1:400; ref. AB9610, Millipore) with mouse monoclonal anti-CC1 (1:100; ref. OP80, Calbiochem) antibodies. Puncta were immunostained with guinea pig anti-VGAT (1:500; ref. 131 004, Synaptic Systems) and mouse anti-γ2 (1:500; ref. 224011; Synaptic Systems) and NG2 cells with rabbit anti-NG2 (1:400; ref AB5320, Millipore). All primary antibodies were washed 3 times in PBS and incubated in secondary antibodies coupled to DyLight-405, Alexa-488, or Alexa-633 for 2 hr at room temperature (1:500; ref. 106-475-003, Jackson ImmunoResearch and ref. A11029 and A21071, Life Technologies, respectively).

Interneurons and NG2 cells were recorded with intracellular solutions containing 5.4 mM biocytin. Slices containing injected cells were fixed overnight in 4% paraformaldehyde at 4°C. For identification of FSIs, interneurons were immunostained with rabbit anti-PV antibody (1:2000; ref. PV-25, Swant) in VGAT-Venus;NG2-DsRed mice. Biocytin was revealed with Cy-5 conjugated streptavidin (ref. 016-170-084, Jackson Immuno- Research) during incubation with the secondary antibody. Negative controls for immunostainings were performed by omitting all primary antibodies or by incubating a primary antibody with a secondary antibody against an omitted primary antibody.

Optical sections of confocal images were sequentially acquired using a 10× or 63× oil objectives (NA = 1.4) with the LSM-710 software (Zeiss). Images were processed and analyzed using ImageJ and Imaris softwares. For counting layer V Olig2$^+$/CC1$^+$ cells, we analyzed 270 × 270 µm of 10–20 Z-sections (each 1 µm). For counting layer V PV$^+$ interneurons, we analyzed 225 × 225 µm of 55 Z-sections (each 0.5 µm) from PV immunostainings of barrel cortex of Venus$^+$ mice. Co-localization of biocytin-loaded NG2 cells and VGAT$^+$/PV puncta as well as NG2$^+$ cells and VGAT$^+$/γ2$^+$ was assessed on 125 × 125 µm 60 Z-sections (each 0.32 µm). 3D surfaces were created for all channels after applying a median filter to reduce noise. First, we quantified the number of VGAT$^+$ puncta per NG2 cell somata and branches. For this, we extracted the fluorescent profiles for VGAT puncta and biocytin-loaded NG2 cells by tracing a line crossing both the puncta and the biocytin-loaded region. We considered as a contacting puncta those showing more than 23% overlapping of fluorescent profiles. Finally, we checked whether or not those VGAT$^+$ puncta on NG2 cells colocalized with PV. Countings were performed for 4 biocytin-loaded NG2 cells and on 6–8 branches per NG2 cell. Similar colocalization parameters were applied for VGAT$^+$/γ2$^+$ on NG2$^+$ cells (n = 8 cells from two mice). To determine the spatial distribution of all VGAT$^+$ and VGAT$^+$/PV$^+$ puncta around biocytin-loaded NG2 cells, we first obtained the 3D coordinates for all puncta (Imaris tools). Then, puncta densities (number of puncta per µm$^3$) were calculated in increasing eccentric volumes from the center of the stack where the NG2 cell soma was located. Each volume corresponded to a sphere to which spherical caps were removed to remain inside the stack (*Figure 2—figure supplement 2B*). The first volume was calculated from a sphere of 5-µm radius around the center of the field. The next volume was calculated from a sphere of 10-µm radius to which we subtracted the previous smaller volume. We repeated this volume calculation until we attained the x, y axis boundaries of the field. For each calculated volume, we determined all VGAT$^+$ and VGAT$^+$/PV$^+$ puncta densities.

## Statistics

Data are expressed as mean ± SEM. The nonparametric Mann–Whitney U test for independent samples was used to determine statistical differences between different pairs. When

comparisons within single pairs were required, the Wilcoxon signed-rank test for related samples was used (GraphPad InStat software version 3.06). Cumulative distributions were compared using Kolmogorov–Smirnov test. Multiple group comparisons were done using one-way Kruskal–Wallis test followed by a Dunn's multiple comparison post-hoc test. Binomial distributions and confidence intervals (Wilson interval; *Brown et al., 2001*) for connection probabilities of FSIs and NFSIs were obtained using a custom routine in Python kindly provided by Christophe Pouzat (*Supplementary file 1*). Correlations were tested with a Pearson r test.

## Acknowledgements

We thank Stéphane Dieudonné for helpful comments on the manuscript, all physicists of the department, in particular E Papagiakoumou that helped us with the holographic system, Christophe Pouzat for statistical advice, and the SCM Imaging Platform of the Saints-Pères Biomedical Sciences site of Paris Descartes University for confocal images. Venus was developed by Dr Atsushi Miyawaki at RIKEN, Wako, Japan. This work was supported by grants from Agence Nationale de la Recherche (ANR), Fondation pour la Recherche sur le Cerveau (FRC), and Fondation pour l'aide à la recherche sur la Sclérose en Plaques (ARSEP). VE thanks the FRC and the Rotary Club for financial support through the program 'Espoir en Tête' 2012. DO, and PPM were supported by fellowships from Nerf-île-de-France and from Ecole des Neurosciences de Paris (ENP), respectively. The MCA team is part of the ENP-Ile-de-France network.

## Additional information

### Funding

| Funder | Grant reference | Author |
|---|---|---|
| Agence Nationale de la Recherche | R14193KK | Maria Cecilia Angulo |
| Federation pour la Recherche sur le Cerveau | R13227KK | Maria Cecilia Angulo |
| Ecole des Neurosciences de Paris | Fellowship | Paloma P Maldonado |
| Nerf ile de France | Fellowship | David Orduz |
| Federation pour la Recherche sur le Cerveau and Rotary Club | Grant | Valentina Emiliani |

The funders had no role in study design, data collection and interpretation, or the decision to submit the work for publication.

### Author contributions

DO, PPM, Conception and design, Acquisition of data, Analysis and interpretation of data, Drafting or revising the article; MB, MV-F, Acquisition of data, Analysis and interpretation of data; VS, Contributed unpublished essential data or reagents; YY, VE, Drafting or revising the article, Contributed unpublished essential data or reagents; MCA, Conception and design, Analysis and interpretation of data, Drafting or revising the article

### Ethics

Animal experimentation: All experiments followed European Union and institutional guidelines for the care and use of laboratory of the INSERM. All of the animals were handled according to approved institutional animal care and use protocols of the University Paris Descartes. The protocol was approved by the Committee on the Ethics of Animal Experiments of the University Paris Descartes (Permit Number: CEEA34.MCA.070.12). Every effort was made to minimize suffering.

## Additional files

**Supplementary file**
• Supplementary file 1. Synaptic coupling probabilities.

---

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
