## [Decision Letter]

Thank you for sending your work entitled “Interneurons and oligodendrocyte progenitors form a structured synaptic network in the developing neocortex” for consideration at *eLife*. Your article has been favorably evaluated by Eve Marder (Senior editor), a Reviewing editor, and three reviewers.

The Reviewing editor and the reviewers discussed their comments before we reached this decision, and the Reviewing editor has assembled the following comments to help you prepare a revised submission.

The paper by Oduz, Maldonado et al. uses several methods to demonstrate a surprising selectivity in functional connections between fast spiking interneurons and NG2 cells. The view of the reviewers is that the work is interesting and potentially suitable for publication in *eLife*. However, you will need to provide additional data to convincingly demonstrate some of the claims, which we believe can be collected quite quickly.

One of the reviewers was concerned that the dataset remains too descriptive for publication in *eLife* and does not yet demonstrate a causal link. This reviewer suggested additional experiments using pharmacological or genetic interventions to address the roles of either the γ2 subunit-containing synapses or the innervation by FSI (vs. NFSI), but our view overall is that these experiments would be beyond the scope of the paper. Therefore, we would ask you to address the following points in a resubmission, along with the minor changes:

1) The glutamate uncaging experiments to map the distribution of INs that synapse onto NG2 cells demonstrate a very transient peak of connectivity between INs and NG2 cells, with a 45% connection at P10, but only 20-25% on P8-9 and P11-13. This brings up a major question: if there is such a difference in total connectivity by varying recording by one day, the authors need to clarify the exact time points that were used for determining IN subtype connection in the earlier experiments. The text says experiments were done in the second postnatal week, which would include poorly and highly connected cells/days. This has implications for understanding whether the difference in connection probability of FSI vs NFSI is affected by the stage of development. For example, the proportion of FSI and NFSI that are connected to NG2 cells on each day, and the region of the NG2 cell that they synapse onto. This can be addressed by re-analyzing the already collected data by age to see if there is any variation across development.

2) For the immunostaining to show the distribution of inputs onto biocytin filled NG2 cells the authors should provide some sort of postsynaptic label, even if it is GABA receptors, to more convincingly show that the presynaptic puncta are likely to be forming connections with the NG2 cells rather than on nearby neurons. Similarly, EM analysis of a subset of filled neurons could convincingly show that the fast spiking inputs are forming bonofide synapses onto NG2 cells with the appropriate labeling of the presynaptic and postynaptic cells, this may not be necessary if more immunostaining analysis is done. This becomes even more relevant in regards to the claims that these synapses are transient and rapidly disappear prior to oligodendrocyte myelination. Similarly, using immunolabeling to look at presynaptic and postsynaptic markers onto the NG2 cells at the three ages described electrophysiologically would provide some insight into how connections are lost just prior to oligodendrocyte differentiation.

3) In addition, for the holographic stimulation to map inputs it would be helpful to know whether the mapping of inputs is greatly altered if shorter latency cut offs are used, especially because the connectivity peaks at 40us. Thus, do the distributions look similar even if the number of inputs decreases? This could also help counter any concerns about inadvertently measuring di-synaptic inputs.

---

## [Author Response]

*One of the reviewers was concerned that the dataset remains too descriptive for publication in* eLife *and does not yet demonstrate a causal link. This reviewer suggested additional experiments using pharmacological or genetic interventions to address the roles of either the γ2 subunit-containing synapses or the innervation by FSI (vs. NFSI), but our view overall is that these experiments would be beyond the scope of the paper. Therefore, we would ask you to address the following points in a resubmission, along with the minor changes*:

*1) The glutamate uncaging experiments to map the distribution of INs that synapse onto NG2 cells demonstrate a very transient peak of connectivity between INs and NG2 cells, with a 45% connection at P10, but only 20-25% on P8-9 and P11-13. This brings up a major question: if there is such a difference in total connectivity by varying recording by one day, the authors need to clarify the exact time points that were used for determining IN subtype connection in the earlier experiments. The text says experiments were done in the second postnatal week, which would include poorly and highly connected cells/days. This has implications for understanding whether the difference in connection probability of FSI vs NFSI is affected by the stage of development. For example, the proportion of FSI and NFSI that are connected to NG2 cells on each day, and the region of the NG2 cell that they synapse onto. This can be addressed by re-analyzing the already collected data by age to see if there is any variation across development*.

The transient connectivity was determined with paired recordings experiments rather than with glutamate uncaging experiments as stated in the previous version. We tested 147 NG2 cells with paired recordings compared to 32 with uncaging. With 20 or more tested NG2 cells per day, paired recordings give us a clearer picture of the connectivity probability with respect to the age.

We realize from the reviewers’ comment that presenting the transient connectivity at the end of the results raises several questions about the experiments described before. Therefore, we included a new Figure 1 with the connectivity probability of FSI and NFSI with respect to the age. We pooled the data obtained at PN8-9, PN10-11 and PN12-13 to increase the number of connected pairs for each of the two groups of interneurons (3 or more for FSI and 6 or more for NFSI for a total tested pairs of 45, 44 and 44 at PN8- 9, PN10-11 and P12-13, respectively). Note the peak of connectivity at PN10-11 for both FSI and NFSI in Figure 1. The transient increase in connectivity occurs for both groups of interneurons at PN10-11 and is not differently affected by the age of the animal (see the last paragraph of the subsection headed “Fast-spiking interneurons are highly connected to NG2 cells”). We also clarify at which ages were performed all the experiments throughout the text (particularly in the Results section). The transient connectivity does not impact electrophysiological properties such as the kinetics of postsynaptic events of FSI-NG2 cell and NFSI-NG2 cell connections (see subsection “FSIs and NFSIs target specific segregated subcellular domains”) and single vs. double release sites (see “Restricted number of release sites per interneuron”).

*2) For the immunostaining to show the distribution of inputs onto biocytin filled NG2 cells the authors should provide some sort of postsynaptic label, even if it is GABA receptors, to more convincingly show that the presynaptic puncta are likely to be forming connections with the NG2 cells rather than on nearby neurons. Similarly, EM analysis of a subset of filled neurons could convincingly show that the fast spiking inputs are forming bonofide synapses onto NG2 cells with the appropriate labeling of the presynaptic and postynaptic cells, this may not be necessary if more immunostaining analysis is done. This becomes even more relevant in regards to the claims that these synapses are transient and rapidly disappear prior to oligodendrocyte myelination. Similarly, using immunolabeling to look at presynaptic and postsynaptic markers onto the NG2 cells at the three ages described electrophysiologically would provide some insight into how connections are lost just prior to oligodendrocyte differentiation*.

The postsynaptic protein assemblies of NG2 cells have not been studied yet and could be substantially different from those found in neurons. For instance, the expression of gephyrin, a major marker of neuronal postsynaptic sites, has not been observed in NG2 cells (Seifi et al., 2014, Front Anat). Therefore, to answer the reviewers’ comment, we used the γ2 subunit of GABA_A_ Rs as a postsynaptic marker since we know from our present and previous studies that this subunit is expressed at many NG2 cell GABAergic synapses (see Figure 3 and [4], Cerebral Cortex). First, we performed triple immunostainings against VGAT, PV (presynaptic markers) and γ2 (postsynaptic marker) in DsRed^+^ cells of NG2-DsRed mice at PN10, the age previously used to analyze VGAT/PV puncta (Figure 2). We observed the presence of VGAT^+^/PV^+^/γ2^+^ puncta on DsRed^+^ cells, but co-localization with the cell was sometimes ambiguous because the DsRed protein is not homogeneously expressed in soma and branches, resulting in cell borders difficult to define. We thus performed triple immunostainings against VGAT (the presynaptic marker), γ2 (the postsynaptic marker) and NG2 (a marker of NG2 cell membranes) in DsRed^+^ cells of NG2-DsRed mice at PN10. Note that NG2 and PV cannot be combined in these experiments since both are rabbit polyclonal antibodies. This is not a real limitation since all considered PV^+^ puncta were also VGAT^+^ puncta (Figure 2). The PV/VGAT colocalization is mandatory since PV can also be expressed outside synapses. New Figure 3 illustrates the presence of numerous VGAT^+^/γ2^+^ synaptic puncta on NG2 ^+^ cell membranes of the soma and branches as previously observed in biocytin-loaded cells (same co-localization parameters; see Materials and methods, subsection headed “Immunostainings and puncta analysis”). In addition, VGAT^+^/γ2^+^ synaptic puncta on neurons can be clearly distinguished from those present on NG2 cells at this developmental stage (new Figure 3). These results confirm that VGAT^+^ puncta analyzed previously are formed on NG2 cells and not on nearby neurons, and provide in addition a protein confirmation of the presence of γ2 at many NG2 cell synapses (see subsection entitled “FSIs and NFSIs target specific segregated subcellular domains ”). Finally, we also examine VGAT^+^/γ2^+^ synaptic puncta in NG2^+^ membranes of DsRed^+^ cells at PN8 and PN13 and, in line with our electrophysiological data showing functional synapses at these stages, we also observed VGAT^+^/ γ2^+^/NG2^+^ co- localization. However, we believe that a reliable study comparing the three ages to provide some insight into how connections are anatomically lost during oligodendrogenesis is technically challenging. The variability of the number of puncta from cell to cell, the small size of a synaptic puncta to observe any possible morphological modification, the lack of a more general postsynaptic marker and the loss of NG2 and DsRed expression when NG2 cells undergo differentiation constitute major technical difficulties to properly address this question. Although challenging, the study of anatomical changes occurring during the transient connectivity would be interesting to do in future studies. Nevertheless, we have already provided some insight into how functional connections are lost in previous reports by combining different electrophysiological, pharmacological and molecular experiments (39; 4).

*3) In addition, for the holographic stimulation to map inputs it would be helpful to know whether the mapping of inputs is greatly altered if shorter latency cut offs are used, especially because the connectivity peaks at 40us. Thus, do the distributions look similar even if the number of inputs decreases? This could also help counter any concerns about inadvertently measuring di-synaptic inputs*.

The GABAergic responses induced in NG2 cells by holographic photo -stimulation of Venus^+^ interneurons cannot be di-synaptic because these neurons are inhibitory (GABA has already a hyperpolarizing effect in neurons in the second postnatal week). Upon photo-activation, interneurons will inhibit rather than excite other surrounding neurons. It is also highly unlikely that our photo-stimulation activates other structures such as dendrites of excitatory neurons which may indirectly activate an interneuron and generate di-synaptic response in NG2 cells. Indeed, when we displaced the light spot outside the targeted interneuron to the neuropil where many dendrites are axons are laying, we got ambiguous responses in only 2.8% of the cases (see subsection entitled “Local spatial arrangement of interneuron-NG2 cell connections”). Finally, we also confirmed the mono-synaptic nature of the connections by patching the targeted presynaptic interneuron and performing paired recordings (Figure 5—figure supplement 3).

However, we agree that this is a critical point for the study. We thus performed the analysis proposed by the reviewers with a cut-off at 50 ms which corresponds to the delay of action potential generation of around 80% of interneurons (Figure 5—figure supplement 3; x-axis labels were changed to better appreciate this proportion). The distributions of connected and unconnected cells were still different for NG2 cells (Kolmogorov-Smirnov test, D=0.2513, P<0.05) . We also compared the new distributions with those reported in the previous manuscript. We found no significant differences of the mean inter-somatic distance of connected interneurons (43.9±2.4 µm *vs* 42.9±3.5 µm; Mann -Whitney U test, P>0.05) or the distribution of inter-somatic distances of connected interneurons between the two different cut-offs (Kolmogorov- Smirnov test, D=0.060, P>0.05). Similarly, no changes were observed in the case of pyramidal cells (P>0.05). We concluded that the mapping of inputs is not altered when shorter latency cut-offs are used. We would prefer to keep our previous analysis because, in our opinion, a 50 ms cut-off will cause an underestimation of the number of connected interneurons.